# AMFuse: Add–Multiply-Based Cross-Modal Fusion Network for Multi-Spectral Semantic Segmentation

**Haijun Liu** **, Fenglei Chen, Zhihong Zeng and Xiaoheng Tan ***

School of Microelectronics and Communication Engineering, Chongqing University, Chongqing 400044, China;
haijun_liu@cqu.edu.cn (H.L.); flyc@cqu.edu.cn (F.C.); azhihong@cqu.edu.cn (Z.Z.)
* Correspondence: txh@cqu.edu.cn

**Abstract:** Multi-spectral semantic segmentation has shown great advantages under poor illumination conditions, especially for remote scene understanding of autonomous vehicles, since the thermal image can provide complementary information for RGB image. However, methods to fuse the information from RGB image and thermal image are still under-explored. In this paper, we propose a simple but effective module, add–multiply fusion (AMFuse) for RGB and thermal information fusion, consisting of two simple math operations—addition and multiplication. The addition operation focuses on extracting cross-modal complementary features, while the multiplication operation concentrates on the cross-modal common features. Moreover, the attention module and atrous spatial pyramid pooling (ASPP) modules are also incorporated into our proposed AMFuse modules, to enhance the multi-scale context information. Finally, in the UNet-style encoder–decoder framework, the ResNet model is adopted as the encoder. As for the decoder part, the multi-scale information obtained from our proposed AMFuse modules is hierarchically merged layer-by-layer to restore the feature map resolution for semantic segmentation. The experiments of RGBT multi-spectral semantic segmentation and salient object detection demonstrate the effectiveness of our proposed AMFuse module for fusing the RGB and thermal information.

**Keywords:** multi-spectral images; cross-modal feature fusion network; semantic segmentation; salient object detection

## 1. Introduction

Semantic segmentation, aiming to predict the semantic category of each pixel in an image, is a fundamental problem of remote sensing [1,2] and multimedia applications [3–5]. For example, it is of critical significance to autonomous vehicles, since exact remote scene segmentation at pixel level can ensure the reliable operation of autonomous vehicles in complicated real-world environments.

With the single visible modality, the admittedly great process has been achieved on semantic segmentation in visible RGB images over the years [6]. However, those approaches only can perform well on images captured with sufficient daylight. Their performance would drastically deteriorate under challenging conditions such as poor lighting conditions, occlusions, or low object resolution. Especially in a 24-h scene, visible cameras only are insufficient for autonomous vehicles, when the light is adverse or unavailable (e.g., during night). The visible cameras cannot capture the appearance information of key objects. In this case, image capturing devices that do not depend on visible light are necessary to improve the performance under such difficult conditions, such as thermal cameras or depth cameras. Recently, depth images are always captured by RGBD cameras (e.g., Kinect), which are rarely deployed in practical surveillance systems because they are expensive, always used indoors and on images with distance limitations. Another exceptionally well-suited modality is thermal imaging; since humans often have a higher temperature compared to the surrounding background, their emitted radiation can be sensed well by thermal

cameras. As shown in Figure 1, the pedestrian in the red rectangle can be sensed by thermal camera while the RGB camera could not capture any appearance information of it. Moreover, thermal cameras are also commonly used in practical video surveillance systems.

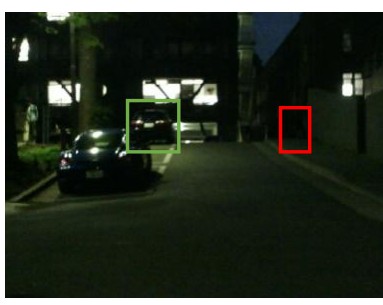 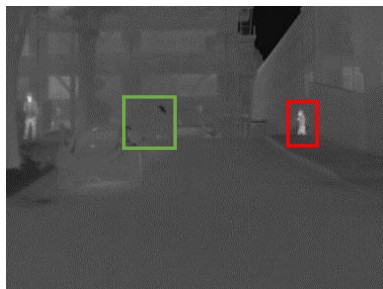

RGB image                    Thermal image

**Figure 1.** Illustration of the aligned multi-spectral images, where the RGB image and thermal image can provide complementary information for each other.

As the thermal cameras become increasingly popular, multi-spectral image analysis is emerging as an active research topic [7–10]. Although a number of previous works have attempted to explore the role of thermal images in semantic segmentation, several issues remain.

**RGB–thermal multi-spectral semantic segmentation are still under-explored.** Despite the increasing amount of research on RGB semantic segmentation that has been published in the recent years [11], there are only a few deep-learning-based works focusing on RGB–thermal multi-spectral semantic segmentation. MFNet [12], RTFNet [13] and AFNet [14] are the three most typical models, which adopt convolutional neural networks for RGB–thermal multi-spectral semantic segmentation, achieving onlygeneral performance. Therefore, there is large room for researchers to further improve the segmentation performance. Moreover, some researchers are trying to explore the role of thermal images combined with RGB images for multi-spectral pedestrian detection [7,8,15,16].

**Effective feature extraction and multi-spectral feature fusion methods are urgent needed.** With RGB and thermal multi-spectral images, finding a way to fuse them is the biggest challenge for semantic segmentation to leverage the complementary information well, as shown in Figure 1. There are two simple strategies to fuse features from different modalities: early fusion [17,18] and late fusion [19]. Early fusion focuses on fusing them on an image level—the input of deep models–while late fusion concentrates on fusing them on a result level—the output of deep backbone models. The two kinds of fusion methods always face challenges in extracting informative multi-spectral features or effectively fusing the results. An increasing number of studies have tended to adopt the middle-fusion strategy [12,13,16,20], which utilizes two models to separately extract the features of different modalities, and then fuses them in the feature level to produce the final results. How to extract the features and fuse them still remains to be explored.

**Motivation.** To tackle the multi-spectral semantic segmentation task, we propose the addition of a multiply fusion (AMFuse) module, consisting of two fundamental math operations (addition and multiplication) for leveraging all the available information existing in one or both modalities. The addition operation focuses on extracting *cross-modal complementary features*; simultaneously, the multiplication operation concentrates on the *cross-modal common features*. Moreover, the attention [21] module and atrous spatial pyramid pooling (ASPP) [22] module are also incorporated into our proposed AMFuse module to enhance the multi-scale context information, since the objects in the scene are always with multiple scales and the number of pixels of each class are extremely imbalanced. In the UNet-style encoder–decoder framework, the ResNet [23] model is adopted as the encoder backbone for feature extraction. Our method adopts the middle-fusion strategy mentioned above. The proposed AMFuse modules fuse the multi-spectral features separately from the two independent RGB and thermal encoders, and then replace the copy

and crop operation in UNet to bridge the encoder and decoder. As for the decoder part, the multi-scale information obtained from our proposed AMFuse modules is hierarchically merged layer-by-layer to restore the feature map resolution for semantic segmentation.

The main contributions can be summarized as follows:

1.  We propose the AMFuse modules, focusing on both the cross-modal complementary features and common features to take advantage of all the information from both modalities.
2.  By incorporating the proposed AMFuse modules into the ResNet-based UNet-style framework, we can achieve superior performance for RGBT multi-spectral semantic segmentation and salient object detection.

## 2. Related Work

This section briefly reviews those related works from two aspects: semantic segmentation of natural images and multi-spectral image analysis.

### 2.1. Semantic Segmentation of Natural Images

Here, we mainly pay attention to the network architectural improvements for semantic segmentation. The pioneering CNN-based semantic segmentation model, fully convolutional networks (FCN) [24], was proposed for pixel-wise labeling. FCN adopts the existing image classification networks, such as VGG [25] and GoogleNet [26], to extract the semantic features, which then are upsampled to the desired resolution through deconvolutional networks. To improve the ability of deconvolutional networks, the concept of encoder–decoder network architecture is proposed for semantic segmentation. The two typical models are SegNet [27] and UNet [28]. SegNet [27] adopts the VGG16 network as the encoder, and the mirrored version as the decoder. Specifically, the decoder uses pooling indices computed in the max-pooling step of the corresponding encoder to perform non-linear upsampling. UNet [28] consists of a contracting path (encoder), an expansive path (decoder) and additional skip-connections to bridge encoder and decoder. The skip-connections can improve the model's accuracy and address the problem of vanishing gradients. Furthermore, several modified versions [29,30] of encoder–decoder networks have been applied to semantic segmentation. DeepLabV3+ [22] has outperformed many state-of-the-art segmentation networks on PASCAL VOC2012 [31] and Cityscapes [32] datasets. It combines the advantages of both dilated convolutions and feature pyramid pooling to build the atrous spatial pyramid pooling (ASPP) module, which can encode multi-scale contextual information by applying atrous (dilated) convolution at multiple scales. GCNet [33] unifies the non-local network and squeeze-excitation network into a three-step general framework for global context modeling. Recently, some researchers introduce the transformers to perform the semantic segmentation task for the state-of-the-art performance [34,35].

### 2.2. Multi-Spectral Image Analysis

There are mainly four kinds of RGB–thermal multi-spectral image analysis applications: cross-modality person re-identification [9,10,36], pedestrian detection [7,8,15,16], semantic segmentation [12–14,37] and salient object detection [38–40].

RGB–thermal cross-modality person re-identification [9,10,36] aims to search for a person of interest from multi-disjoint cameras deployed at different locations, where the query image may be obtained from the thermal camera during nighttime, while the gallery images may be captured by the RGB camera during the daytime. The challenge lies in the large cross-modal discrepancy and large intra-modal variations. RGB–thermal pedestrian detection [7,8,15,16] extends the RGB pedestrian detection by incorporating the aligned thermal image to address the adverse illumination conditions, occlusions and clutter background. Similar to multi-spectral semantic segmentation, the key point is how to take advantage of the complementary information from the multi-spectral images.

As for the multi-spectral semantic segmentation, there are three related works—FNet [12], RTFNet [13] and AFNet [14]—for urban scenes. MFNet [12] constructs the first RGBT multi-spectral semantic segmentation dataset, and designs two identical encoders for RGB and

thermal images, respectively. The encoder consists of some proposed mini-inception blocks with dilated convolutions. Moreover, a short-cut block was designed for fusing the feature maps from the RGB and thermal encoders by concatenation operation. RTFNet [13] is still in the encoder–decoder framework, adopting the ResNet model as encoder for informative feature extraction and designing a new decoder (Upception block) to restore the feature map resolution. AFNet [14] introduces a co-attention mechanism by designing an attention fusion module to calculate the spatial correlation between the RGB image and thermal image feature maps, and to guide the fusion of features from different spectra.

RGBT salient object detection [38–40] aims to find the object that human eyes pay much attention to in an aligned RGB and thermal infrared image pair. Wang et al. [38] collected the first RGBT-salient object detection dataset and proposed a multi-task manifold ranking algorithm. Then, Tu et al. [39] contributed a large-scale dataset and comprehensive benchmark for RGBT salient object detection, which aggregates multi-level multi-modal features with attention mechanism, and shows great improvement against the previous methods. Furthermore, Tu et al. [40] proposed a more suitable network with multi-interaction Siamese decoder to utilize the multi-type cues in a reasonable way and take the modalities bias into account simultaneously.

Moreover, some other RGB-D (RGB and Depth)-images-based semantic segmentation and salient object detection methods [20,41–43] are also related to our work. The only difference is that the additional input image is thermal image or depth image. Finding a way to fuse them is a common core research point.

## 3. Method

In this section, we introduce the framework of our proposed add–multiply-based cross-modal fusion network (AMFuse) for multi-spectral semantic segmentation, as depicted in Figure 2. The framework is in line with the UNet style, mainly consisting of two components—(1) Encoder–decoder: two independent DownConv (downsampling + convolution) encoders, respectively, for RGB and thermal images, and one UpConv (upsampling + convolution) decoder for semantic segmentation to restore the desired resolution. (2) Our proposed AMFuse modules for middle-level cross-modal feature fusion to bridge the encoder and decoder.

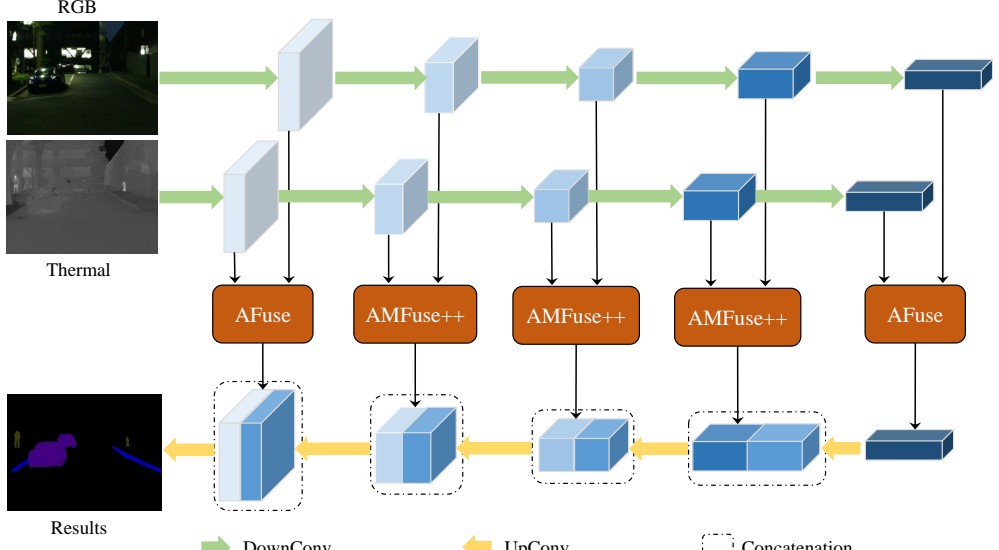

**Figure 2.** The outline of our framework for multi-spectral semantic segmentation, which mainly consists of two components: the encoder–decoder UNet-style framework and the proposed AMFuse modules. The encoder–decoder UNet-style framework includes two independent DownConv (downsampling+convolution) encoders, respectively, for RGB and thermal images, and one UpConv (upsampling+convolution) decoder for semantic segmentation to restore the desired resolution. Our

proposed AMFuse modules are hierarchically merged layer-by-layer to fuse the cross-modal RGB–thermal features for bridging the encoder and decoder. There are five positions (*Pos*) to utilize the fusion modules. At the beginning and the end of the encoders for fusing the RGB–thermal features (*Pos*1 and *Pos*5), we utilize two AFuse modules, while the other three places (*Pos*2, *Pos*3 and *Pos*4) are AMFuse++ modules. The detailed structures of AMFuse series modules are described in Section 3.2.

### 3.1. Encoder–Decoder

As the UNet architecture has been confirmed as an effective framework in many semantic segmentation networks for natural images and medical images, we also adopt the encoder–decoder UNet-style structure.

Since RGB and thermal images are from different modalities, we claim that they should be separately processed to extract the complementary information from each other. As carried out in [12,13], we design two encoders to, respectively, extract features of RGB and thermal images. For simplicity in the following fusion modules, the two encoders have identical structures to each other. One point we should pay attention to is that the number of input channels in the first layer of the two encoders is different, since RGB image is three channels and thermal image is only one channel in the multi-spectral images.

We preferentially employ the ResNet [23] model to be the encoder backbone, due to its excellent performance for feature learning as well as its relatively concise architecture. The ResNet model mainly consists of one shallow convolution block and four res-convolution blocks—in total, five DownConvs in each encoder, as shown by the green in Figure 2. The shallow convolution block sequentially includes a convolutional layer, a batch normalization layer and a ReLU activation layer. Following the shallow convolution block, a max pooling layer and four res-convolution blocks are sequentially employed to gradually reduce the resolution and increase the number of channels of the feature maps. The details of the ResNet model can refer to [23].

As for the decoder part, we do not design a mirrored version of the decoder as carried out in SegNet [27]. We adopt the simplest way to build the decoder to collaborate with encoders. There are also five UpConvs (yellow arrows in Figure 2). Each UpConv only sequentially consists of a upsampling layer and a convolutional layer. The results of UpConv and fusion module are concatenated, and hierarchically merged layer-by-layer to restore the feature map resolution for semantic segmentation, as carried out by Unet [28].

### 3.2. AMFuse Module

Given two feature maps, RGB $X_v \in R^{h \times w \times c}$ and thermal $X_t \in R^{h \times w \times c}$, the goal of fusion module is to obtain the fused feature maps $X_f \in R^{h \times w \times c}$ with the same feature size.

Since the two feature maps $X_v$ and $X_t$ are from different modalities, with different focuses of the scene objects and containing complementary information for each other, we argue that there are two aspects should be well addressed.

1. **The cross-modal complementary information.** There are some objects only can be sensed by one camera. For example, as shown in Figure 1, during the nighttime one pedestrian only appears in the thermal image (red rectangle), while one car only has indistinct appearance in the RGB image (green rectangle). Methods to fuse these two feature maps to take advantage of those complementary information existing in only one modality is a key problem.
2. **The cross-modal common information.** There are some objects can be sensed by two cameras, which should also be well processed during fusion to enhance them instead of weakening one against the other.

To address the two problems, we adopt two simple math operations: element-wise addition ($\oplus$) for extracting the cross-modal complementary information and element-wise multiplication ($\otimes$) for extracting the cross-modal common information.

**AFuse:**

$$X_f = X_v \oplus X_t. \tag{1}$$

**MFuse:**

$$X_f = X_v \otimes X_t. \tag{2}$$

AFuse equally treats the RGB feature and thermal feature to integrate the cross-modal complementary information through one element-wise addition operation, while Mfuse equally treats the RGB feature and thermal feature to integrate the cross-modal common information through one element-wise multiplication operation.

As the toy example in Figure 3, 0 means that the object is unsensed by the camera, while 1 and 2 denote that the object is sensed by the camera. The element-wise addition ($\oplus$) operation can filter out the cross-modal complementary information $1 = 1 + 0 = 0 + 1$ or $2 = 1 + 1$ (the object is sensed by at least one camera); otherwise, $0 = 0 + 0$. The element-wise multiplication ($\otimes$) can filter out the common information, only when $1 = 1 * 1$ (the object is simultaneously sensed by two cameras); otherwise, $0 = 1 * 0 = 0 * 1$ or $0 = 0 * 0$.

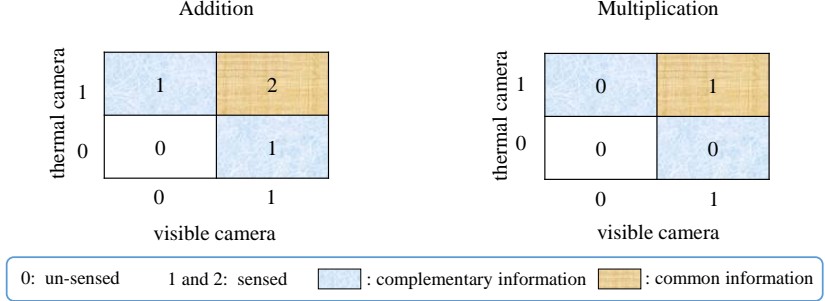

**Figure 3.** The illustration of element-wise addition ($\oplus$) for extracting the cross-modal complementary information and element-wise multiplication ($\otimes$) for extracting the cross-modal common information.

After addressing the cross-modal complementary information and common information by addition and multiplication operations, respectively, we should design methods to simultaneously integrate them. Here, we consider two widely used approaches, addition and concatenation operations.

The first one involves directly adding the results of AFuse and MFuse modules, which can be termed as **AMFuse**,

$$X_f = X_v \oplus X_t \oplus (X_v \otimes X_t). \tag{3}$$

The latter involves directly concatenating (**Cat**) the results of AFuse and MFuse modules, and then reducing the channel number through a $1 \times 1$ convolutions (**Conv**), which can be termed as **AMFuse+**,

$$X_f = \mathbf{Conv}\big(\mathbf{Cat}(X_v \oplus X_t, X_v \otimes X_t)\big). \tag{4}$$

Figure 4 depicts the detail structures of our proposed **AFuse**, **MFuse** modules, and their combinations, **AMFuse** and **AMFuse+**.

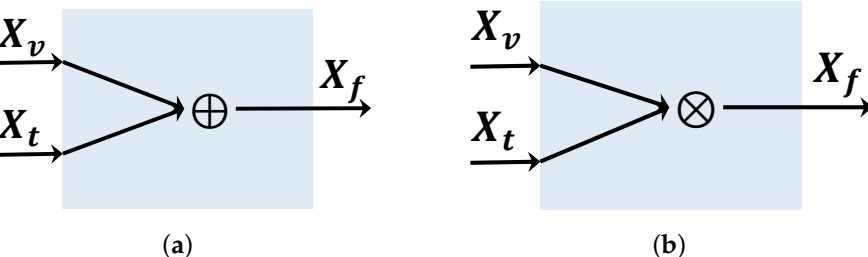

**Figure 4.** *Cont.*

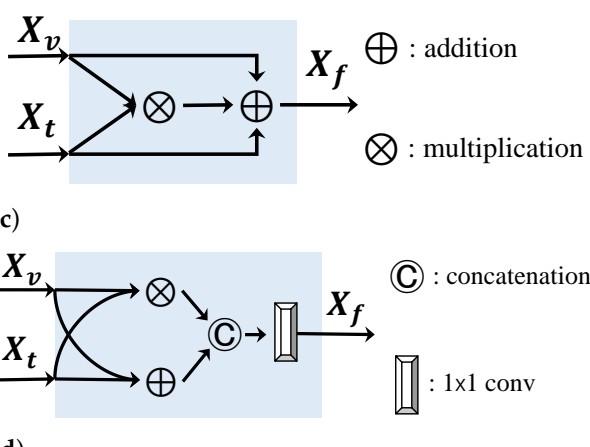

**Figure 4.** The illustration of AMFuse series modules, the addition fusion, multiplication fusion and their combinations. (**a**) AFuse; (**b**) MFuse; (**c**) AMFuse; (**d**) AMFuse+.

Moreover, due to the commonly existing challenges in scene semantic segmentation, such as one object with multi scales and different classes with extremely imbalance of pixels numbers, we try to introduce the attention module [21] and atrous spatial pyramid pooling (ASPP) module [22] to enhance the multi-scale context information. The attention module [21] can efficiently suppress distractors to improve the features for focusing on objects, while the ASPP module [22] can integrate all the coarse and fine features from multiple scales. Based on the aforementioned **AMFuse+** module, we only empirically incorporate the attention module (**A**), respectively, for RGB feature $X_v$ and thermal feature $X_t$ to perform the addition operation. Then the results of **Conv** are processed by **ASPP** module to address the multi-scale property. We term it as **AMFuse++**,

$$X_f = \textbf{ASPP}\Big\{\textbf{Conv}\Big(\textbf{Cat}\big(\textbf{A}(X_v) \oplus \textbf{A}(X_t), X_v \otimes X_t\big)\Big)\Big\}. \tag{5}$$

A detailed structure of **AMFuse++** is shown in Figure 5, where the attention and ASPP modules are shown in Figures 6 and 7, respectively.

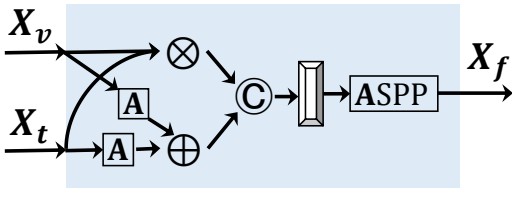

AMFuse++

**Figure 5.** The illustration of AMFuse++ module, where **A** is the attention module and **ASPP** is the atrous spatial pyramid pooling module.

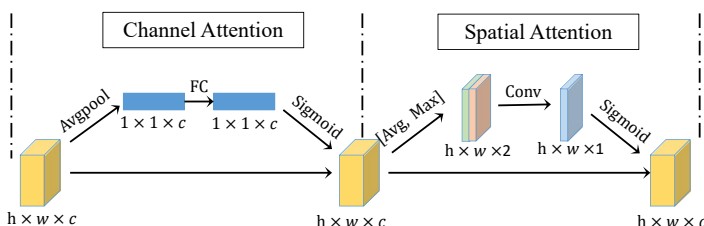

**Figure 6.** The structure of attention module [21], sequentially consisting of channel attention and spatial attention.

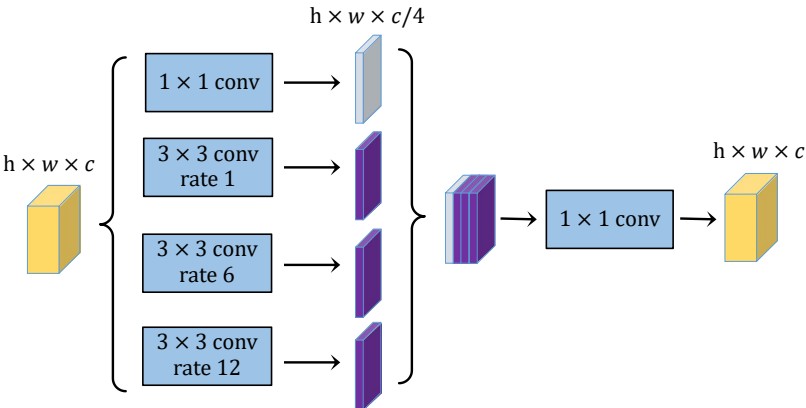

**Figure 7.** The structure of ASPP module with deep separable convolutions proposed in Deeplab v3+ [22].

*3.3. Loss Function*

The segmentation prediction is generated by a simple head with a $1 \times 1$ convolution layer, which directly translates the input feature maps to $M$ maps, where $M$ is the number of classes. The full network is trained end-to-end with the binary cross entropy loss $L_{bce}$ and dice loss $L_{dice}$.

$$L = L_{bce} + L_{dice}, \qquad (6)$$

where $L_{bce} = -\sum_i [G_i \log(P_i) + (1 - G_i) \log(1 - P_i)]$ and $L_{dice} = 1 - \frac{2|G \cap P|}{|G| + |P|}$, $G$ and $P$ are the ground truth and predictions, $i$ denotes the pixel index, $|\cdot|$ is the number of elements in a set. $L_{bce}$ and $L_{dice}$ provide effective local (pixel-level) and global (image-level) supervision for accurate segmentation.

## 4. Results

In this section, we evaluate the effectiveness of our proposed AMFuse serial modules for multi-spectral semantic segmentation on a public urban scene dataset.

*4.1. Experimental Settings*

**Dataset.** We adopt the public RGB–thermal segmentation dataset released in [12]. This utilized an InfRec500 camera to record the urban scenes, which can simultaneously capture the RGB and thermal information. The dataset includes 15,569 aligned RGB–thermal pairs with the resolution of $480 \times 640$, among which 749 taken at nighttime and 820 taken at daytime. There are nine hand-labeled semantic classes in the ground truth, including one background class and eight classes of obstacles commonly encountered during driving (pedestrian, car, bike, curve, color cone, guardrail, car stop and bump). We follow the dataset splits of [12] to conduct the experiments. According to the time series of image capturing, the training set includes 50% of the daytime images and 50% of the nighttime images, whereas the validation and testing sets contain 25% of the daytime images and 25% of the nighttime images, respectively.

**Evaluation metrics.** Following the settings of [12,13], we also adopted two metrics to quantitatively evaluate the segmentation performance. One is the accuracy (Acc) for each class, which is also known as recall, and the other one is the intersection over union (IoU) for each class. The average values across all the classes for the two metrics are denoted as mAcc and mIoU, respectively, which can be calculated as follows:

$$mAcc = \frac{1}{M} \sum_{i=1}^{M} \frac{TP_i}{TP_i + FN_i}, \qquad (7)$$

$$mIoU = \frac{1}{M} \sum_{i=1}^{M} \frac{TP_i}{TP_i + FP_i + FN_i}, \qquad (8)$$

where $M$ is the number of classes. Different from [12], we set $M = 9$ including the unlabeled background classes. $TP$, $FP$ and $FN$ denote the true positive, false positive and false negative computed on the ground truth $G$ and predictions $P$, respectively.

**Implementation details.** The implementation (https://github.com/FlyC235/AMFuse, accesssed on 7 June 2022) of our method is with the Pytorch framework. The ResNet model is adopted as the encoder backbone network, and the pre-trained ImageNet parameters are adopted for the network initialization. We adopt the stochastic gradient descent (SGD) optimizer for optimization, and the momentum and weight decay parameter are set to 0.9 and 0.0005, respectively. We set the initial learning rate as 0.01, and exponential decay strategy with $\gamma = 0.95$ adopted to gradually decrease the learning rate. Moreover, the training data are augmented using the flip technique. Specially, to accelerate the training, we adjust the input size of images in every five epochs, as follows: $240 \times 320$ for the first three epochs, sequentially following $480 \times 640$ for the left two epochs.

### 4.2. Comparisons to State-of-the-Art

In this section, we compare our proposed methods to some state-of-the-art semantic segmentation methods, including some models designed for three-channel RGB images (ERFNet [44], PSPNet [45], SegNet [27] and DUC-HDC [46]) and four kinds of fusion models for two modalities (FuseNet [43], MFNet [12], RTFNet [13] and AFNet [14]). For fair comparison, the input layers of those models designed for three-channel RGB images are modified to accommodate the four-channel RGB–thermal images. For the model architecture of our AMFuse serial method, we fixedly utilize two AFuse modules at the beginning and the end of encoders to fuse RGB and thermal information, and three AMFuse++ modules for another three places, as shown in Figure 2.

#### 4.2.1. The Overall Performance

The accuracy (Acc) and intersection over union (IoU) results for each class (we only list the results for those eight classes of obstacles, since the unlabeled pixels occupy most of the images, always obtaining similar segmentation results across different methods with a little information), and the average values across all the classes (mACC and mIoU) on the testing dataset are listed in Table 1, from which we can see the following:

1.  In general, our proposed AMFuse serial methods perform much better compared to those existing semantic segmentation methods in terms of both mAcc and mIoU metrics.

2.  Compared to the state-of-the-art method (RTFNet [13]) for RGB–thermal multi-spectral semantic segmentation, our proposed AMFuse methods obtain superior performance. When compared to AMFuse-18 to RTFNet-152: mACC (64.1 vs. 63.1), mIoU (53.1 vs. 53.2), AMFuse-18 performs a little better. However, our method only utilizes resnet-18 as backbone of encoders, while RTFNet utilizes resnet-152 with much more parameters. It demonstrates the effectiveness of our proposed AMFuse module for fusing the RGB and thermal information.

3.  By comparing the AMFuse methods with different backbones, we can see that AMFuse-50 performs better than both AMFuse-18 and AMFuse-152. However, when modifying the AMFuse++ module by adding a $1 \times 1$ convolution layer for the multiplication operation, AMFuse-152* outperforms AMFuse-50 and AMFuse-152. We conjecture the reason is that the $1 \times 1$ convolution can refine the common information extracted by the multiplication operation from the multi-spectral data.

4.  The results for different classes are with very big differences in terms of Acc and IoU values, which is caused by the extremely unbalanced distribution of classes in the dataset [12]. In general, for each class the less the number of pixels is, the worse the result is. Especially for Guardrail class, the results may be 0.0, since Guardrail class occupies the fewest pixels, resulting in insufficient training for it. Moreover, there are only 4 images containing Guardrail class among 393 images in the testing dataset.

**Table 1.** The comparisons to the state-of-the-art methods (%). Our proposed AMFuse methods and RTFNet method are with different ResNet backbone encoders, such as ResNet-18, ResNet-50 and ResNet-152.

| Methods | Car | | Pedestrian | | Bike | | Curve | | Car Stop | | Guardrail | | Color Cone | | Bump | | mAcc | mIoU |
|---|---|---|---|---|---|---|---|---|---|---|---|---|---|---|---|---|---|---|
| | Acc | IoU | Acc | IoU | Acc | IoU | Acc | IoU | Acc | IoU | Acc | IoU | Acc | IoU | Acc | IoU | | |
| ERFNet [44] | 78.8 | 667.1 | 62.9 | 56.2 | 41.6 | 34.3 | 39.4 | 30.6 | 12.6 | 9.4 | 0.0 | 0.0 | 0.1 | 0.1 | 33.0 | 30.5 | 40.8 | 36.1 |
| PSPNet [45] | 81.0 | 74.8 | 69.2 | 61.3 | 63.8 | 50.2 | 44.7 | 38.4 | 18.1 | 15.8 | 0.0 | 0.0 | 36.4 | 33.2 | 49.0 | 44.4 | 51.3 | 46.1 |
| SegNet [27] | 67.5 | 65.3 | 60.3 | 55.7 | 61.0 | 51.1 | 46.3 | 38.4 | 10.4 | 10.0 | 0.0 | 0.0 | 41.9 | 12.0 | 55.3 | 51.5 | 49.1 | 42.3 |
| DUC-HDC [46] | 91.5 | 84.8 | 76.4 | 68.8 | 66.7 | 54.6 | 54.7 | 41.9 | 30.9 | 19.2 | 12.3 | 4.4 | 40.2 | 34.3 | 61.5 | 45.1 | 59.3 | 50.1 |
| FuseNet [43] | 81.0 | 75.6 | 75.2 | 66.3 | 64.5 | 51.9 | 51.0 | 37.8 | 17.4 | 15.0 | 0.0 | 0.0 | 31.1 | 21.4 | 51.9 | 45.0 | 52.4 | 45.6 |
| MFNet [12] | 77.2 | 65.9 | 67.0 | 58.9 | 53.9 | 42.9 | 36.2 | 29.9 | 12.5 | 9.9 | 0.1 | 0.0 | 30.3 | 25.2 | 30.0 | 27.7 | 45.1 | 39.7 |
| RTFNet-50 [13] | 91.3 | 86.3 | 78.2 | 67.8 | 71.5 | 58.2 | 59.8 | 43.7 | 32.1 | 24.3 | 13.4 | 3.6 | 40.4 | 26.0 | 73.5 | 57.2 | 62.2 | 51.7 |
| RTFNet-152 [13] | 93.0 | 87.4 | 79.3 | 70.3 | 76.8 | 62.7 | 60.7 | 45.3 | 38.5 | 29.8 | 0.0 | 0.0 | 45.5 | 29.1 | 74.7 | 55.7 | 63.1 | 53.2 |
| AFNet [14] | 91.2 | 86.0 | 76.3 | 67.4 | 72.8 | 62.0 | 49.8 | 43.0 | 35.3 | 28.9 | **24.5** | **4.6** | 50.1 | 44.9 | 61.0 | 56.6 | 62.2 | 54.6 |
| AMFuse-18 (ours) | 90.2 | 84.2 | 81.9 | 70.7 | 76.1 | 60.9 | 58.0 | 42.6 | 34.3 | 26.5 | 20.7 | 3.0 | 53.4 | 43.4 | 62.8 | 48.9 | 64.1 | 53.1 |
| AMFuse-50 (ours) | 91.0 | 86.7 | 82.9 | 72.7 | 75.3 | 61.5 | **61.2** | 46.2 | 37.8 | 29.2 | 22.3 | 4.2 | 53.3 | 46.7 | 64.6 | 54.0 | 65.3 | 55.5 |
| AMFuse-152 (ours) | **94.2** | **88.7** | 82.8 | 72.6 | 78.3 | **63.9** | 57.2 | 45.0 | 31.1 | 25.7 | 11.7 | 1.5 | **57.1** | **48.7** | 71.9 | 52.4 | 64.8 | 55.2 |
| AMFuse-152 * (ours) | **94.2** | **88.7** | **83.1** | **73.0** | **78.6** | 63.1 | 58.4 | **46.5** | **38.9** | **30.1** | 18.0 | 2.9 | 55.3 | 46.9 | **76.3** | **56.7** | **66.9** | **56.2** |

* We modify the AMFuse++ module to $X_f = \mathbf{ASPP}\Big\{\mathbf{Conv}\Big(\mathbf{Cat}\big(\mathbf{A}(X_v) \oplus \mathbf{A}(X_t), \mathbf{Conv}(X_v \otimes X_t)\big)\Big)\Big\}$ by adding a $1 \times 1$ convolution layer for the multiplication operation.

### 4.2.2. Daytime and Nighttime Results

The results of those methods performing on the daytime and nighttime images are also reported in Table 2. We can see that (1) compared to other methods (in columns), our proposed AMFuse methods can achieve the best performance in both the two scenarios. (2) For each method (in rows), the results of nighttime are always better than those of daytime. We suggest that the major reason is the slight misalignment between RGB and thermal images. The spatial misalignment may be from the camera calibration errors and cropping steps, while the temporal misalignment may be from the synchronization errors since RGB camera and thermal camera are always with different frame rates.

**Table 2.** The comparative results (%) on the daytime and nighttime scenarios. * denotes adding a $1 \times 1$ convolution layer for the multiplication operation in the AMFuse++ module.

| Methods | Daytime | | Nighttime | |
|---|---|---|---|---|
| | mAcc | mIoU | mAcc | mIoU |
| ERFNet [44] | 37.5 | 32.5 | 39.3 | 34.5 |
| PSPNet [45] | 42.6 | 37.8 | 49.7 | 45.2 |
| SegNet [27] | 39.9 | 34.6 | 47.4 | 41.7 |
| DUC-HDC [46] | 56.7 | 44.3 | 55.0 | 49.4 |
| FuseNet [43] | 49.5 | 41.0 | 48.9 | 43.9 |
| MFNet [12] | 42.6 | 36.1 | 41.4 | 36.8 |
| RTFNet-50 [13] | 57.3 | 44.4 | 59.4 | 52.0 |
| RTFNet-152 [13] | 60.0 | 45.8 | 60.7 | 54.8 |
| AFNet [14] | 54.5 | 48.1 | 60.2 | 53.8 |
| AMFuse-18 (ours) | 58.2 | 46.2 | 61.5 | 53.1 |
| AMFuse-50 (ours) | 60.6 | **49.0** | 61.7 | 54.5 |
| AMFuse-152 (ours) | 60.7 | 48.2 | 61.7 | 55.0 |
| AMFuse-152 * (ours) | **61.3** | 48.9 | **63.6** | **55.8** |

In the daytime, both the RGB and thermal cameras can informatively capture images with good visual quality. Therefore, the slight misalignments between RGB and thermal images would confuse the model training, leading to the performance degrading. However, at the nighttime, the RGB camera would sense few objects almost with all black in the images, while the thermal camera can still work well as in the daytime. In this case, the thermal information will dominate the model prediction, so the slight misalignments have little impact on the segmentation performance.

### 4.2.3. Qualitative Demonstrations

Figure 8 displays the sample qualitative demonstrations of multi-spectral semantic segmentation in urban scenes with typical daytime and nighttime scenarios. We can see that (1) RGB and thermal images truly could provide complementary information for each other. For example, in the second, column though the pedestrian is invisible in RGB image, those methods still can segment the pedestrian with the help of thermal image. (2) Our proposed AMFuse serial methods perform better than the current state-of-the-art method RTFNet [13], especially for those little objects. For instance, RTFNet would lose some small objects (the little pedestrian in column 1, the little color cone in column 5 and the little bike in column 6, as depicted in the red circles), while our methods can perceive them. This maybe benefit from the addition and multiplication operations for fusing RGB and thermal information, as well as the attention and ASPP modules incorporation for enhancing the multi-scale context information.

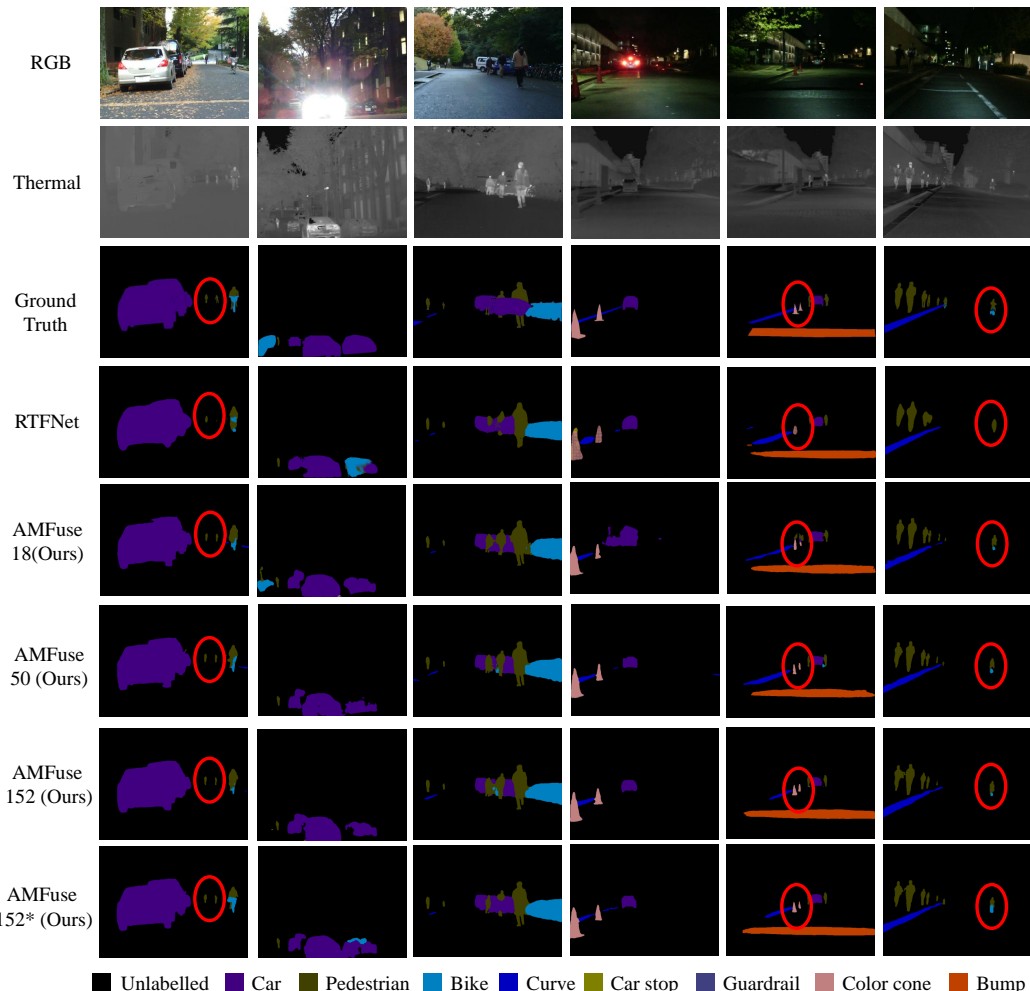

**Figure 8.** The sample qualitative demonstrations of multi-spectral semantic segmentation in the urban scenes. The left three and right three columns display the segmentation results with typical lighting conditions in the daytime and nighttime, respectively. The figure is best viewed in color.

## 5. Discussion

In this section, first, we evaluate the effectiveness of our proposed AMFuse modules from the following aspects. Then, to verify the generalization ability of our proposed AMFuse models, we applied it to the RGBT SOD task. Finally, the improvements were compared to previous research of the proposed methods are discussed.

*5.1. Ablation Study*

5.1.1. The Effectiveness of ResNet as Backbone of Encoder

Here, we adopt the ResNet-18 model as the backbone of encoders, comparing to the original UNet [28]. We set two cases: one is for four-channel (4c) inputs with one encoder, while the other is, respectively, for 1-channel thermal inputs and three-channel RGB inputs with two independent encoders utilizing the AFuse module for feature fusion. As listed in Table 3, ResNet-18 backbone performs much better than UNet in both two cases. The AFuse methods with two independent encoders outperform the four-channel inputs with one encoder methods, indicating the effectiveness of two-stream networks, respectively, processing RGB and thermal images compared to one-stream network simultaneously processing them.

**Table 3.** The effectiveness of ResNet-18 as the backbone of encoder (%).

| Methods | mAcc | mIoU |
|:---:|:---:|:---:|
| UNet (4c) | 46.7 | 41.2 |
| UNet + AFuse | 47.1 | 41.4 |
| ResNet-18 (4c) | 59.3 | 50.6 |
| ResNet-18 + AFuse | 61.9 | **52.4** |
| ResNet-18 (RGB) | **62.3** | 49.2 |
| ResNet-18 (Thermal) | 57.3 | 46.0 |

Moreover, in Table 3, we also list the results of models with only one encoder for RGB and thermal images, respectively. The only RGB inputs perform much better than the only thermal inputs. However, when fusing the RGB and thermal images with AFuse module, the performance is greatly improved especially in terms of mIoU metric.

5.1.2. The Effectiveness of Mixing AFuse and AMFuse++ Modules

As shown in Figure 2, there are five positions to utilize the fusion modules. For the model architecture of our AMFuse serial method, we empirically utilize two AFuse modules at the beginning and end of encoders to fuse RGB and thermal information (*Pos*1 and *Pos*5), and AMFuse++ modules for another three places (*Pos*2, *Pos*3 and *Pos*4), denoted as shown in method ⑤. To experimentally demonstrate the effectiveness of this setting, we additionally conducted two kinds of experiments with ResNet-18 and ResNet-50 backbones: (1) using only AFuse (method ①) or only AMFuse++ (method ④) modules; (2) fixing the AMFuse++ modules at *Pos*2, *Pos*3 and *Pos*4, and changing the modules at *Pos*1 and *Pos*5 to be AMFuse (method ②) and AMFuse+ (method ③). The results are listed in Table 4. First, we can see that our method ⑤ mixing AFuse and AMFuse++ modules outperforms methods ① and ④, especially under the mIoU metric. Second, compared to method ⑤, methods ②, ③ and ④ perform worse, which demonstrates the effectiveness of AFuse modules at *Pos*1 and *Pos*5. In our opinion, the first fusion module is with sufficient details and high resolutions, while the final fusion module is with high-level semantic information and low resolutions. Therefore, to preserve the information originating from the backbone, we empirically fix 2 AFuse modules for *Pos*1 and *Pos*5, while we only fuse the other levels with our AMFuse++ module. Third, as to different backbones, we can see that AMFuse-50 performs much better than AMFuse-18 in all settings.

**Table 4.** The effectiveness of mixing AFuse and AMFuse++ modules (%).

| Method | *Pos*1 | *Pos*2 | *Pos*3 | *Pos*4 | *Pos*5 | Backbone | mAcc | mIoU |
|--------|--------|--------|--------|--------|--------|----------|------|------|
| ① | AFuse | AFuse | AFuse | AFuse | AFuse | ResNet-18 | 61.9 | 52.4 |
|   |       |       |       |       |       | ResNet-50 | 64.5 | 53.2 |
| ② | AMFuse | AMFuse++ | AMFuse++ | AMFuse++ | AMFuse | ResNet-18 | 62.2 | 52.0 |
|   |        |          |          |          |        | ResNet-50 | 64.5 | 54.6 |
| ③ | AMFuse+ | AMFuse++ | AMFuse++ | AMFuse++ | AMFuse+ | ResNet-18 | 62.0 | 51.6 |
|   |         |          |          |          |         | ResNet-50 | 64.7 | 54.9 |
| ④ | AMFuse++ | AMFuse++ | AMFuse++ | AMFuse++ | AMFuse++ | ResNet-18 | 61.1 | 52.3 |
|   |          |          |          |          |          | ResNet-50 | 65.1 | 54.8 |
| ⑤ | AFuse | AMFuse++ | AMFuse++ | AMFuse++ | AFuse | ResNet-18 | 64.1 | 53.1 |
|   |       |          |          |          |       | ResNet-50 | **65.3** | **55.5** |

### 5.1.3. The Effectiveness of Attention and ASPP Modules

Based on the aforementioned ResNet-18+AFuse model, we sequentially added attention and ASPP modules. Table 5 lists the corresponding results. We can see that, when solely incorporating one module, only one metric value can be improved (mIoU for attention, mAcc for ASPP). However, when incorporating the two modules simultaneously, two metric values can be improved. It demonstrates the effectiveness of attention and ASPP modules for enhancing the multi-scale context information.

**Table 5.** The effectiveness of attention (Atten) and ASPP modules (%).

| Methods | mAcc | mIoU |
|---------|------|------|
| Baseline (ResNet-18 + AFuse) | 61.9 | 52.4 |
| Baseline + Atten | 61.4 | 52.7 |
| Baseline + ASPP | 62.5 | 51.2 |
| Baseline + Atten + ASPP | **63.4** | **53.1** |

### 5.1.4. The Effectiveness of AMFuse Serial Modules

Finally, in this subsection we evaluate our proposed AMFuse serial modules (described in Section 3.2 and Figures 4 and 5). The whole architecture framework is illustrated in Figure 2, we fixed two AFuse modules to the beginning and the end of encoders to fuse RGB and thermal information, and only adjusted the fusion modules in the left three middle places, sequentially with AFuse, MFuse, AMFuse, AMFuse+, AMFuse++ and AMFuse++* modules. AMFuse++* denotes that we utilized the attention modules in both of the addition and multiplication branches. The ResNet-50 model is adopted as the backbone of encoders. Table 6 lists the corresponding results, from which we see that:

1. When fusing the RGB and thermal information with only one operation (addition (AFuse) or multiplication (MFuse)), MFuse performs much worse than AFuse. It is intuitively reasonable, since we argue that the addition operation could focus on the cross-modal complementary components while the multiplication operation could concentrate on the cross-modal common components. Those cross-modal complementary information existing in the RGB and thermal features would dominate the multi-spectral semantic segmentation task. The detail results for each class are listed in Table 7, and some sample qualitative demonstrations are shown in Figure 9.

2. When fusing the RGB and thermal information simultaneously with addition and multiplication operation (AMFuse with addition operation and AMFuse+ with concatenation operation), the performance is greatly improved. It demonstrates effectiveness of our method to address the cross-modal complementary information and cross-modal common information simultaneously. AMFuse+ slightly outperforms AMFuse, indicating that compared to the addition operation, the concatenation operation maybe could give more freedoms for feature refining with multiple channels.

3. Based on AMFuse+, AMFuse++ can achieve large improvements in terms of the mIoU metric. It demonstrates the effectiveness of incorporating the attention and ASPP modules for enhancing the multi-scale context information, especially for those small objects, as shown in Figure 8.
4. AMFuse++* performs worse than AMFuse++, denoting that applying the attention module to both the addition and multiplication branches is worse compared with the one only applying to addition branch. The reason may be that the common information simultaneously existed in two modalities is sensitive to multiplication operation, which is easy distorted by the attention operation separately performed on two modalities.
5. Compared to AFuse, MFuse, AMFuse and AMFuse+ methods, AMFuse++ introducing attention and ASPP modules are with more additional parameters and FLOPs. However, AMFuse++ achieved large improvements in terms of the mIoU metric.

**Table 6.** The effectiveness of AMFuse serial modules based on the ResNet-50 backbone encoders.

| Methods | mAcc (%) | mIoU (%) | Para (M) | FLOPs (G) |
|---|---|---|---|---|
| ResNet-50 + AFuse | 64.5 | 53.2 | 109.3 | 67.1 |
| ResNet-50 + MFuse | 62.9 | 52.5 | 109.3 | 67.1 |
| ResNet-50 + AMFuse | 65.1 | 54.3 | 109.3 | 67.1 |
| ResNet-50 + AMFuse+ | **65.3** | 54.5 | 120.4 | 69.9 |
| ResNet-50 + AMFuse++ | **65.3** | **55.5** | 149.5 | 96.3 |
| ResNet-50 + AMFuse++ * | 64.6 | 53.7 | 149.5 | 96.3 |

* denotes that we utilized the attention modules in both of the addition and multiplication branches, $X_f = \mathbf{ASPP}\Big\{ \mathbf{Conv}\Big( \mathbf{Cat}\big( \mathbf{A}(X_v) \oplus \mathbf{A}(X_t), (\mathbf{A}(X_v) \otimes \mathbf{A}(X_t)) \big) \Big) \Big\}$.

**Table 7.** The comparisons of MFuse, AFuse and AMFuse modules based on ResNet-50 backbone (%).

| Methods | Car | | Pedestrian | | Bike | | Curve | | Car Stop | | Guardrail | | Color Cone | | Bump | | mAcc | mIoU |
|---|---|---|---|---|---|---|---|---|---|---|---|---|---|---|---|---|---|---|
| | Acc | IoU | Acc | IoU | Acc | IoU | Acc | IoU | Acc | IoU | Acc | IoU | Acc | IoU | Acc | IoU | | |
| MFuse | 89.2 | 83.5 | 79.8 | 70.8 | 75.1 | 60.2 | 56.2 | 41.5 | 33.3 | 24.5 | 24.1 | 3.3 | 50.2 | 44.3 | 59.9 | 46.2 | 62.9 | 52.5 |
| AFuse | **90.6** | 84.2 | 79.2 | 70.6 | 73.9 | 60.3 | 56.0 | 41.5 | 38.9 | **28.0** | 24.5 | **5.1** | 55.1 | 46.1 | **63.6** | 47.4 | 64.5 | 53.2 |
| AMFuse | 90.2 | **84.9** | 81.7 | 71.6 | 75.1 | 60.6 | 57.1 | 42.0 | 39.0 | 28.0 | 27.3 | 5.1 | 55.8 | 49.9 | 61.4 | **49.5** | 65.1 | 54.3 |

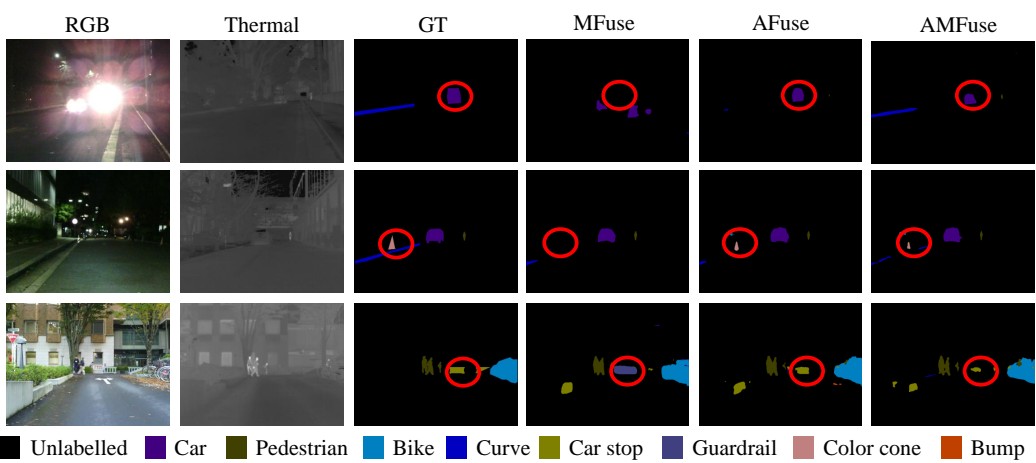

**Figure 9.** The sample qualitative demonstrations of MFuse, AFuse and AMFuse modules based on ResNet-50 backbone.

## 5.2. Application to RGBT-Salient Object Detection

RGBT-salient object detection (SOD) aims to estimate the common conspicuous objects or regions in an aligned visible and thermal infrared image pair, similar to our RGBT multi-spectral semantic segmentation. They both take similar manners to manage the RGB and thermal infrared data. The only difference is that RGBT semantic segmentation predicts

multiple semantic classes, while RGBT SOD only predicts two classes, one for salient object and the other for background. Therefore, to verify the generalization ability of our proposed AMFuse models, we can also extend it to perform the RGBT SOD task.

### 5.2.1. Dataset

We adopt a large-scale public RGBT SOD dataset, VT5000, recently released in [39]. VT5000 includes 5000 aligned RGBT image pairs, and has complex scenes and various objects to cover all the challenging problems in RGBT SOD, including multiple salient object, low illumination, center bias, thermal crossover, image clutter, big salient object, small salient object, cross image boundary, out of focus, similar appearance and bad weather. Following the setup of [39,40], we choose 2500 various images in VT5000 for training, and the rest 2500 image pairs are taken as the testset.

### 5.2.2. Experimental Setup

Following the setup of [40], we adopt the stochastic gradient descent (SGD) optimizer for optimization, and the momentum and weight decay parameter are set to 0.9 and 0.0005, respectively. We set the initial learning rate as $1 \times 10^{-3}$, and decrease it to $1 \times 10^{-4}$ after 20 epochs and $1 \times 10^{-5}$ after 50 epochs. Following the settings of [39,40], we also adopt five widely used metrics to quantitatively evaluate the SOD performance, including F-measure ($Fm$), weighted F-measure $wF$, S-measure $Sm$, E-measure $Em$, and mean absolute error $MAE$.

We compare our method with some existing methods, including 3 traditional RGBT SOD methods, SDGL [47], MTMR [38], M3S-NIR [48]; 2 deep learning based RGBT SOD methods, ADF [39], SiamDecoder [40]; and 2 deep learning based RGBD SOD methods, DMRA [49], S2MA [50].

### 5.2.3. Comparison Results

Table 8 lists the results of our AMFuse serial methods and other state-of-the-art methods on the VT5000 testing dataset.

1.  The three traditional methods perform much worse than those deep learning based methods, demonstrating the effectiveness of deep learning for feature extraction.
2.  For the two RGBD SOD methods, DMRA [49] and S2MA [50], they performs poor on the RGBT SOD dataset. The reason maybe lie in the nature of RGBD and RGBT SOD tasks. In RGBD SOD task, the depth channel is always adopted as auxiliary information, while in RGBT SOD task, the RGB and thermal modalities are with equivalent importance for extracting the complementary and common information.
3.  Our AMFuse serial methods perform well against the existing RGBT SOD methods under the above 5 metrics, which all focus on the fusion of RGB and thermal information. The superiority of our methods demonstrates the effectiveness of the proposed AMFuse modules for fusing the RGB and thermal information.

**Table 8.** The comparisons to the state-of-the-art methods on VT5000 RGBT SOD dateset. Our proposed AMFuse methods are with different ResNet backbone encoders, such as ResNet-18, ResNet-50 and ResNet-152. * denotes adding a $1 \times 1$ convolution layer for the multiplication operation in the AMFuse++ module.

| Methods | *Em* | *Sm* | *Fm* | *MAE* | *wF* |
|---------|------|------|------|-------|------|
| MTMR [38] | 0.795 | 0.680 | 0.595 | 0.114 | 0.397 |
| M3S-NIR [48] | 0.780 | 0.652 | 0.575 | 0.168 | 0.327 |
| SDGL [47] | 0.824 | 0.750 | 0.672 | 0.089 | 0.559 |
| DMRA [49] | 0.696 | 0.672 | 0.562 | 0.195 | 0.532 |
| S2MA [50] | 0.869 | 0.855 | 0.751 | 0.055 | 0.734 |

**Table 8.** *Cont.*

| Methods | Em | Sm | Fm | MAE | wF |
|---|---|---|---|---|---|
| ADF [39] | 0.891 | 0.864 | 0.778 | 0.048 | 0.722 |
| SiamDecoder [40] | 0.897 | 0.868 | 0.801 | 0.043 | 0.763 |
| AMFuse-18 (ours) | 0.893 | 0.852 | 0.785 | 0.043 | 0.753 |
| AMFuse-50 (ours) | 0.903 | 0.867 | 0.802 | **0.039** | **0.784** |
| AMFuse-152 * (ours) | **0.918** | **0.872** | **0.823** | **0.039** | 0.751 |

Figure 10 shows some qualitative results. We can see that our methods can well locate all the salient objects. Our AMFuse modules equally treat the RGB and thermal modality, and fuse the features with addition and multiplication operations to take advantage of the cross-modality complementary and common information. However, there are still some unsatisfied cases, especially for some images with semantic ambiguity and obscure boundary. For example, in the 2nd column, the little object in the red circle is with semantic ambiguity. It is hard to say whether it is a salient object. In the 4th column, the car is invisible in the RGB image, while the boundary of the car in the thermal image is obscure.

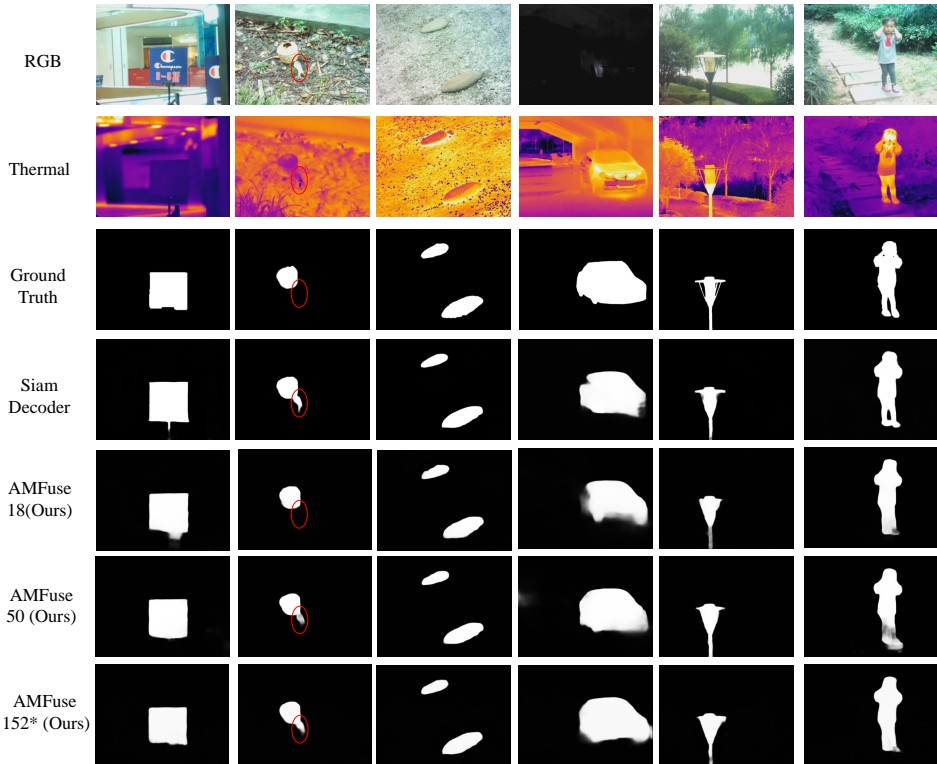

**Figure 10.** The sample qualitative demonstrations of RGBT multi-spectral SOD. We select 6 RGBT image pairs with diverse challenges to compare the quality of the predicted SOD maps.

## 6. Conclusions

We presented add–multiply fusion (AMFuse) modules in this paper, which can improve the semantic segmentation performance for RGBT multi-spectral remote scene images. The main idea is that we simultaneously focus on both the cross-modal complementary features and common features to take advantage of all the information from both modalities. Moreover, to enhance the multi-scale context information, we incorporate the attention and ASPP modules into our AMFuse module. In the ResNet-based UNet-style framework, AMFuse modules fuse the multi-spectral features from two encoders, and then are hierarchically merged to bridge the encoders and decoder. Experiments on a multi-spectral semantic segmentation dataset in urban scenes and a RGBT SOD dataset show that AMFuse

module can obviously improve the performance. It demonstrates the effectiveness and superiority of our proposed AMFuse method.

Although achieving a relative fine fusion of multi-spectral features, the proposed AMFuse method still has room for improvement. At present, influenced by the backbone, our AMFuse module still occupies a relatively high computational cost. There should be more effective ways to design the AMFuse module. Therefore, our future work will focus on the further optimization of the fusion method of RGB and thermal features, reducing the complexity of the AMFuse module while maintaining its performance, and introducing an efficient transformer mechanism to improve the fusion method and replace the ResNet backbone.

**Author Contributions:** Conceptualization, H.L.; methodology, F.C. and H.L.; validation, F.C. and Z.Z.; writing—original draft preparation, H.L. and F.C.; writing—review and editing, H.L. and X.T. All authors have read and agreed to the published version of the manuscript.

**Funding:** This work was supported in part by the National Natural Science Foundation of China under Grants 62001063, U20A20157 and U2133211, and in part by the China Postdoctoral Science Foundation under Grant 2020M673135, Chongqing Postdoctoral Research Program under Grant XmT2020050.

**Data Availability Statement:** https://www.mi.t.u-tokyo.ac.jp/static/projects/mil_multispectral/ (accessed on 10 July 2022).

**Conflicts of Interest:** The authors declare no conflict of interest.

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
