# Peer review of "AMFuse: Add–Multiply-Based Cross-Modal Fusion Network for Multi-Spectral Semantic Segmentation"

_remotesensing, doi:10.3390/rs14143368_

Round 1

Reviewer 1 Report

The manuscript presents a module for RGB and thermal information fusion, called AMFuse. The proposed technique has been described in detail in Section 3 while the state-of-the-art techniques have been briefly described in Section 2. In my opinion, it is necessary to clearly justify the novelty of the proposed technique with respect to the state-of-the-art approaches. The fundamental of AMFuse seems to be very simple: additions to obtain complementary information and products to use complementary information. Are there other approaches that use this kind of operations? What are the differences?

The authors have compared AMFuse with a wide range of state-of-the-art techniques (most of them are very recent). The results show that the proposed technique obtain significant improvements respect the other techniques. I think it would be interesting to include an analysis of the execution time required by the different techniques. In practical applications, this aspect can be critical.

I suggest the authors include some additional explanations about the sample qualitative demonstrations (such as Fig. 8).

There are some minor errors (see, for example, line 195).

Reviewer 2 Report

The paper introduces a method for performing semantic segmentation of RGBT data (RGB images and thermal images). The input is a pair of images (RGB and thermal). The images are processed independently in the encoding branch of a U-net architecture and fused and concatenated to the decoding branch of the U-net architecture.
Results obtained by this architecture on a data-set are presented and compared to related approaches. 

An ablation study is used to highlight the impact of the different design choices for the network. 

The term “achieving general performance” on lines 45-46 is not clear. 

Line 59 “still remain exploration” (“under exploration”?) 

“imabalance” is not a verb (line 68)

Line 112 “lies in”

Line 143 “is inline with” (“in line”) 

In the caption of Fig. 2, the terms “posX” (where X=1, …, 5) are not defined. (The numbering could be from left to right or right to left)

Line 174, both Xv and Xt have the same dimensions (it shouldn’t be the case).  

Line 175 Xf has the same dimensions as Xv and Xt (it shouldn’t be the case). 

Line 175 “with the same feature size” as what? 

Line 178 “that should be addressed”

I don’t know if it is necessary to repeat “math operations” all the time. It seems fine to just use the term “operations”. 

The meaning of the paragraph between the lines 190 and 193 is not clear.
How can element-wise addition and multiplication be done in (1) and (2) given that Xv and Xt have different numbers of channels (3 for Xv and 1 for Xt). Are you broadcasting? 

The sentence between lines 193 and 195 is incomplete (“As that the object”)

Are you performing any sort of normalization/rescaling of the values in Xt? It contains thermal information (expressed in degree Celsius or degree Fahrenheit) while Xv contains RGB information (likely with real values in the unit interval)

What does it mean to “empirically incorporate” a module on p. 6? 

Is there any reason why the attention module is only applied for the arguments of the addition in (5)? 

Why do you use binary cross entropy if you have multiple classes? (p. 7) Do you allow one pixel to belong to multiple classes? 

In Table 1, what is the reasoning behind adding a convolution for the product? Why isn’t it discussed in Section 3? 

Point 2 on p. 9, AMFuse-18 is compared to RTFNet-512, the conclusion reached is that AMFuse gives better results and uses a lighter module. An inspection of the results in Table 1 shows that RTFNet-512 works very poorly on one category (‘Guardrail’), which leads to worse results on average. Based on the reasons given in [13] (and repeated in this paper), it seems reasonable to discard this class for the analysis. In that case, the advantage of AMFuse-18 is not evident anymore. 

“Qualitative demonstrations” (instead of “The qualitative demonstrations”) as the title for the section 4.2.3. (same for section 4.2.2)

In Fig. 7 why are the cars occluding the pedestrian in the ground truth images? 

In table 4 why is there a double line between (4) and (5)? 

Line 329 “in my opinion”: I thought that the paper had four authors? 

Reviewer 3 Report

Some comments:

a. Explain the novelty of the research work.

b. Why this manuscript should be published in Remote Sensing, a mdpi journal.

c. Which is the main innovation of tha manuscript?

d. The main objective of the manuscript is not clear and well-defined in the manuscript.

e. A discussion section of the research work must be added in the revised version of the manuscript.

f. Methodology is not very clear. It must be improved. Please, add a figure in which Authors can explain the whole process.

g. In section 5. Conclusions section is very naive. Authors must revised in a profound way including main outcomes of the research, principal conclusions achieved and also limitations of the work.

h. English must be revised.

i. Please, improve a state-of-the-art in terms of the area research work.

j. In my opinion, reconsideration after major revision. Thank you.

Round 2

Reviewer 1 Report

The authors have followed most of my suggestions.

Reviewer 3 Report

I have no additional comments.